# Dual-Drug Loaded Separable Microneedles for Efficient Rheumatoid Arthritis Therapy

**DOI:** 10.3390/pharmaceutics14071518

**Published:** 2022-07-21

**Authors:** Mengchen An, Mengxiao Shi, Jingjing Su, Yueru Wei, Rongrong Luo, Pengchao Sun, Yongxing Zhao

**Affiliations:** 1Department of Pharmaceutics, School of Pharmaceutical Sciences, Zhengzhou University, No. 100 Science Ave, Zhengzhou 450001, China; amc@gs.zzu.edu.cn (M.A.); 202022342015559@gs.zzu.edu.cn (M.S.); 202011341030341@gs.zzu.edu.cn (J.S.); 202012342015413@gs.zzu.edu.cn (Y.W.); 202012342015411@gs.zzu.edu.cn (R.L.); 2Key Laboratory of Targeting Therapy and Diagnosis for Critical Diseases, Zhengzhou University, No. 100 Science Ave, Zhengzhou 450001, China; 3Key Laboratory of Advanced Pharmaceutical Technology, Ministry of Education of China, Zhengzhou University, Zhengzhou 450001, China

**Keywords:** separable microneedles, tocilizumab, aptamer Apt1-67, rheumatoid arthritis

## Abstract

Although the inhibitors of the interleukin-6 receptor (IL-6R) and tumor necrosis factor-α (TNF-α) have achieved a certain success in the clinical treatment of rheumatoid arthritis (RA), great effort should be made to overcome side effects and to improve patient compliance. The present research aimed to address these problems by the co-delivery of tocilizumab (TCZ)—an inhibitor of IL-6R—and an aptamer Apt1-67, which specifically inhibits TNF receptor 1 via separable microneedles (MN). MN were featured with a sustained release of TCZ from needle tips and a rapid release of Apt1-67 from needle bodies by using methacrylate groups grafted hyaluronic acid as the fillings of needle tips and polyvinyl alcohol/polyvinyl pyrrolidone as the fillings of needle bodies. Our results demonstrated that TCZ and Apt1-67 were distributed in MN as expected, and they could be released to the surroundings in the skin. In vivo studies revealed that combined medication via MN (TCZ/Apt1-67@MN) was superior to MN loaded with a single drug. Compared with subcutaneous injection, TCZ/Apt1-67@MN was of great advantage in inhibiting bone erosion and alleviating symptoms of CIA mice. This study not only provides a novel approach for combined medication with different release properties but also supplies a strategy for improving drug efficacy.

## 1. Introduction

Rheumatoid arthritis (RA) is a systemic autoimmune disease characterized by chronic, peripheral, and destructive joint changes [1]. The pathogenesis of RA is still unclear, but it is generally believed that both environmental and genetic factors contribute to the development of RA [2,3]. RA usually elevates the levels of cytokines and chemokines which aggravate the progress of RA in turn [4]. Various drugs, including non-steroidal anti-inflammatory drugs, glucocorticoids, synthetic disease-modifying antirheumatic drugs (DMARDs), and biological DMARDs have been developed and utilized in clinic to relieve the conditions of RA patients [5]. Compared with the conventional synthetic DMARDs, biological DMARDs exhibit a higher therapeutic index with better efficacy [6]. As a typical representative of biological DMARDs, Tocilizumab (TCZ) has been a success for the therapy of RA in clinic by specifically inhibiting the receptors of cytokine interleukin-6 (IL-6R) [7]. TCZ is often administrated intravenously and subcutaneously (SC) [8,9,10]. However, both routes of administration cause pain to patients, and they reduce patient compliance. Therefore, much effort should be made toward the efficient delivery of TCZ, with improved patient compliance.

Microneedles (MN) are an emerging transdermal drug delivery system, and they have been utilized for drug delivery and disease diagnosis [11]. Apart from bypassing the first-pass effect, MN also show advantages over traditional transdermal drug delivery systems with efficient drug penetration [12,13]. MN maintain the characteristics of SC injection, while being painless and easy to self-administer, which can greatly increase patient compliance [14,15]. With a reasonable design of the fillings, stimuli-responsive MN have been developed for the better control of drug release [16,17,18,19]. Except for loading a single drug for monotherapy [20], MN can load multiple drugs to develop a combined therapy system as well [21]. Therefore, MN have a high potential to be a promising drug delivery system that can be used as monotherapy or combination therapy, with improved patient compliance for the treatment of RA.

Tumor necrosis factor-α (TNF-α) is a typical cytokine of inflammation, which specifically binds to TNF receptors (TNFR), including TNFR1 and TNFR2. Binding to TNFR1 induces inflammation and cell apoptosis [22], while activating TNFR2 mainly regulates inflammation and immune responses [23]. At present, various inhibitors of TNF-α have been developed, including mutants of TNF-α, antibodies, and small molecules for RA therapy [24,25]. However, the existing TNF-α inhibitors fail to selectively interact with TNFR1, resulting in inhibiting TNFR2 and inducing serious side effects [22]. To minimize the side effects, inhibitors that only specifically bind to TNFR1 should be developed. For example, a DNA aptamer with 67 bases in length, Apt1-67, can specifically bind to TNFR1 with a high affinity, meanwhile Apt1-67 exhibits good stability [26]. Taking the complex pathogenesis of RA into consideration, preventing TNFR1 and IL-6R at the same time should be an effective strategy for RA therapy.

In the present work, we aimed to develop separable MN as the co-delivery system of TCZ and Apt1-67 for efficient RA treatment with high patient compliance. We first designed a mold with a novel texture and then used a micromolding method to prepare the separable MN. Slow-dissolving mHA (methacrylate groups grafted hyaluronic acid) and fast-dissolving PVA/PVP (polyvinyl alcohol/polyvinyl pyrrolidone) were selected as the fillings of needle tips and needle bodies, respectively. The prepared MN had a sustained release of TCZ from the needle tips and a rapid release of Apt1-67 from the needle bodies. In vivo studies demonstrated that the developed dual-drug loaded MN showed a better anti-RA activity, especially on inhibiting bone erosion and relieving the symptoms of joint cavity of the CIA mice, indicating the great potential for RA treatment.

## 2. Materials and Methods

### 2.1. Materials

Polydimethylsiloxane (PDMS) mold was purchased from Taizhou Microchip Pharmaceutical Technology Co., Ltd. Hyaluronic acid (HA, MW300 KDa) was purchased from Bloomage Biotechnology Co., Ltd. (Jinan, China). Methacrylic anhydride (MA) was obtained from Sigma-Aldrich (Saint Louis, MO, USA). Polyvinyl alcohol (PVA 1788) was donated generously by Jiangxi Alpha Hi-Tech Pharmaceutical Co., Ltd. Polyvinyl pyrrolidone (PVP K88-96) and fluorescein isothiocyanate (FITC) were supplied by Aladdin Biochemical Technology Co., Ltd. (Shanghai, China). Irgacure 2959 (I 2959) and incomplete Freund’s adjuvant were obtained from Sigma-Aldrich (MO, USA). Universal fluorescent dye for hydrogel-Red (EFL-DYE-UF-ENE-R) was obtained from EFL-Tech Co., Ltd. (Suzhou, China). Albumin bovine V (BSA) was purchased from Biotopped Life Science (Beijing, China). Rhodamine B was obtained from Solarbio (Beijing, China). The TCZ injection was obtained from Roche Pharma Ltd. (Basel, Switzerland). Apt1-67 and Apt1-67-Cy5 were synthesized by Huajin Biotechnology Co., Ltd. (Shanghai, China).

Immunization-grade chick type II collagen and complete Freund’s adjuvant (CFA) were purchased from Chondrex, Inc. (Woodinville, MA, USA). An IL-6 enzyme-linked immunosorbent assay (ELISA) kit and a TNF-α ELISA kit were purchased from CUSABIO Biotech (Wuhan, China). All the reagents were used as received without any modification.

### 2.2. Synthesis of Methacrylate Modified Hyaluronic Acid (mHA)

By grafting methacrylate groups on HA through an esterification reaction, mHA was synthesized [27]. Briefly, HA (0.5 g) was mixed with H_2_O (25 mL) and then stirred continuously overnight at 4 °C to obtain a transparent solution, followed by the addition of the co-solvent N,N-dimethylformamide (25 mL). Subsequently, the MA (0.6 mL) solution was introduced, and NaOH (2M) was applied to maintain the pH value of 8~9 for the mixture. The reaction was kept at 4 °C for 24 h. Afterwards, the mixture was precipitated with acetone, washed three times with ethanol, and dialyzed with H_2_O for 48 h to remove any excess MA and organic reagents. The resulting product was dried by lyophilization to obtain mHA. The chemical structure and the grafting rate of mHA were then characterized with ^1^H nuclear magnetic resonance (^1^H NMR, AVANCE III HD 400 M, Karlsruhe, Germany) and Fourier transform infrared spectroscopy (FTIR, Nicolet 2000, Waltham, MA, USA).

### 2.3. Fabrication of the Separable MN

The separable MN were fabricated by the micromolding method with three steps. Firstly, 100 μL of mHA solution (40 mg/mL) was used as the filling to prepare the needle tip of the MN. The filling was poured into a PDMS mold to fill the tips of the MN. The PDMS mold was vacuumed (−60 KPa, 10 min) and centrifuged (25 °C, 5000 rpm, 10 min) to allow the mHA solution to enter the bottom of the PDMS mold cavities. After the removal of the residual mHA solution on the surface, mHA carrying the PDMS mold was centrifuged for another 30 min to ensure that the mHA solution completely accessed the tips. The loading process was repeated three times. Secondly, 200 μL of PVA/PVP solution (25%) was poured into the PDMS mold to prepare the needle body of MN. Similarly, the filled PDMS mold was vacuumed and centrifuged. Thirdly, the base layer was prepared by introducing the PVA/PVP solution (25%) to the PDMS mold. After centrifugation and drying (40 °C, 24 h), the raw MN were peeled off from the PDMS mold and cured with UV irradiation for 30 s.

### 2.4. Characterization of MN

The morphology of the MN was observed with an optical microscope (Leica DM 3000 Leica, Frankfurt, Germany) and a scanning electron microscope (SEM, ZEISS GeminiSEM 300, Oberkochen, Germany) [28]. MN were coated with a thin layer of gold for SEM imaging. The distribution of mHA were visualized with a fluorescent dye (EFL-DYE-UF-ENE-R) that could specifically bind to mHA via a double bond, followed by imaging with a confocal laser scanning microscopy (CLSM, Nikon A1 serious, Tokyo, Japan). The separability of MN was investigated by rinsing the corner of MN with water several times.

The mechanical strength of drug-free MN (UV-curing 0 s, 30 s, 60 s, 120 s) was analyzed with a texture analyzer (TA.HD PlusC, Godalming, UK), using a P5 probe [29]. The trigger force and the test speed were set to 5 g and 0.1 mm/s, respectively. The tested length was 1 mm in total.

To assess the skin penetration ability of the prepared MN, the dorsal of a healthy mouse was depilated, and MN was pressed into the skin for 5 min. The mouse was immediately euthanized, then the application area was stained with trypan blue (0.5%, *w*/*v*, 10 min) [30] and imaged with a camera. Subsequently, MN inserted skin was removed from the body, fixed with tissue fixation, embedded in paraffin, sectioned, stained with hematoxylin and eosin (H & E), and imaged with a microscopy. The post-treatment recovery of the skin was evaluated on the depilated mice. The mouse was fixed on the operating table, and MN were inserted into the skin of the mouse for 10 min. After removal of MN, the MN-treated area was imaged every 30 min until the skin was fully recovered [31]. 

### 2.5. Release of the Model Drug from MN

BSA-FITC was used as a model drug to investigate the effect of the UV curing time on the drug release property from the needle tip of MN. After irradiation by UV light for various periods (0 s, 30 s, 60 s, and 120 s), MN were immersed in 20 mL of PBS and then incubated on a shaker (37 °C, 100 rpm); 1 mL of PBS was removed at the set time points (20 min, 40 min, 1 h, 2 h, 4 h, 8 h, 12 h, 24 h, 48 h, and 72 h), and the same volume of fresh PBS was supplemented. The fluorescence intensity of the released BSA-FITC was determined, and the corresponding concentration was calculated from the calibration curve of BSA-FITC (FI = 32.12C + 11.85, R^2^ = 0.9971).

### 2.6. Drug Loading and Release

TCZ and Apt1-67 were loaded in the needle tips and the needle bodies of MN for the potential anti-inflammatory study. Before drug loading, the effect of UV irradiation on the stability of TCZ and Apt1-67 were investigated. TCZ solution (0.2 mg/mL) was irradiated with UV light for 0 s, 30 s, 60 s, and 120 s, respectively. Then, the zeta potential of TCZ was measured by a Zetasizer (Nano ZS90, Malvern, UK). Circular dichroism (CD) spectroscopy is widely used to examine the secondary structure of proteins [32]. Therefore, the secondary structure of TCZ was analyzed with a CD spectrometer (Brighttime Chirascan, Leatherhead, UK). Agarose gel electrophoresis was used to verify whether UV irradiation affected the stability of Apt1-67. After UV irradiation (0~120 s), Apt1-67 (10 μM, 5 μL) was mixed with 1 μL 6×loading buffer and loaded to an agrose gel (3%). The gel was run at 100 V for 1 h on an ice bath. Subsequently, the gel was stained with EB (10 mg/mL) and imaged with a Bio-rad gel imaging system (Drive, Hercules, CA, USA).

To load the drugs, TCZ was mixed with mHA at a concentration of 15 mg/mL and Apt1-67 was introduced to the solution of 25% PVA/PVP, with a final concentration of 10 μM to prepare the fillings of the needle tip and the needle body, respectively. Dual drug or single drug loaded MN were constructed as the same method in Section 2.3. The resulting MN were defined as TCZ/Apt1-67@MN, TCZ@MN, and Apt1-67@MN, respectively. FITC labeled TCZ and cy5 functionalized Apt1-67 were used to prepare fluorescent MN to visualize the distribution of the drugs throughout MN by CLSM. After dissolving the fluorescent MN in PBS, the fluorescence intensity of the solution was determined and the drug loading capacity was calculated from the calibration curves of TCZ-FITC (FI = 44.46C − 21.19, R^2^ = 0.9981) and Apt1-67-cy5 (FI = 1346C + 71.22, R^2^ =0.9962), respectively. The fluorescent MN were later applied to the skin of the healthy mouse. After 2 h, MN were removed and the mouse was immediately euthanized. MN-treated skin was separated from the body, and three-dimensional (3D) images of the skin were taken with CLSM to investigate the ex vivo drug distribution.

### 2.7. Cell Culture and Animals

Murine fibroblasts (L929) and human fibroblasts (FLS) cells were cultured at the standard conditions (humidity 100%, 37 °C, and 5% CO_2_), with 1640 and DMEM media plus 10% of fetal bovine serum, respectively. SPF Kunming mice (SCXK (Yu) 2017-0001) were obtained from Laboratory Animal Center of Henan province (Zhengzhou, China) and DBA/1 mice (SCXK (Su) 2016-0010) were purchased from Cavens (Changzhou, China). All animals were raised in an SPF-grade environment. All animal care and experiments were approved by the Animal Ethics Committee of Zhengzhou University, in accordance with the requirements of the National Laboratory Animal Use Law (China).

### 2.8. Cytotoxicity of MN

The cytotoxicity of MN against L929 and FLS cells was evaluated via CCK8 assay [33]. In brief, L929 or FLS cells were seeded in 96 well plates at a density of 5000 cells/well. After incubation at the standard conditions for 24 h, the medium was removed and replaced with the fresh cell culture medium containing different concentrations of MN, which was prepared by dissolving MN in PBS, followed by dilution with RPMI 1640 or DMEM medium. The cells were incubated for another 24 h, and the cell viability was analyzed by CCK8 assay. The optical density (*OD*) at 450 nm of each well was determined with a microplate reader (EnSpire, Waltham, MA, USA), and the cell viability was calculated with the following equation:(1)$$\mathrm{Cell}\;\mathrm{viability}\;\left(\%\right)=\frac{OD_{MN}-OD_{blank}}{OD_{control}-OD_{blank}}\times 100\%$$
where *OD_MN_* represents the *OD* value of the cells treated with MN, *OD_blank_* refers to the *OD* value of the medium, and *OD_control_* is the *OD* value of the cells without treatment.

### 2.9. Anti-RA Activity of MN In Vivo

The collagen-induced arthritis (CIA) model, the most commonly used mouse model for RA study [34], was established to evaluate the anti-inflammatory activity of drug loaded MN in vivo. Chicken type II collagen (5 mL, 2 mg/mL) was added dropwise to CFA (5 mL, 4 mg/mL), with continuous stirring by a homogenizer (IKA, Staufen, Germany) on ice bath. The resulting emulsion (100 μL) was injected subcutaneously to 7-week-old DBA/1 male mice. The second injection was administrated on the 21st day to strengthen the morbidity. The development of arthritis was monitored by the clinical scores, body weight, and paw thickness of the mice. After the establishment of the CIA model, the mice were randomly divided into five groups, and the first four groups were treated with TCZ/Apt1-67@MN, TCZ@MN, Apt1-67@MN, TCZ, and Apt1-67 by subcutaneous injection at the same dose of MN. MN were slipped into the skin overnight to ensure the maximum drug release. The five groups were injected with the same volume of saline as the models (Model). The healthy mice were injected with the same volume of saline as the controls (Control). All the mice were treated every two days. After 10 administrations, the mice were sacrificed, and the levels of TNF-α and IL-6 in serum were determined with ELISA kits. 

After 30 days of decalcification, the paws of mice in each group were embedded in paraffin, and the sections of paws were prepared, followed by staining with H & E and imaging to observe the inflammatory erosion and infiltration of the joints and their pathological changes. In addition, the ankles and the paws of the mice were scanned by Micro-CT (Bruker Skyscan 1276, Karlsruhe, Germany), and the reconstructed 3D images were used to visually observe the paws of mice after treatment.

### 2.10. Statistical Analysis

All data in this experiment are expressed as mean ± standard deviation. Statistical significance was analyzed by one-way ANOVA. *p* value < 0.05 was considered to be statistically significant.

## 3. Results and Discussion

### 3.1. Fabrication and Morphological Characterization of the Separable MN

To prepare the dual-drug loaded separable MN, we first designed a master mold with a novel texture. As seen in Appendix A, a cylindrical needle body (diameter: 350 μm, height: 450 μm) and a conical needle tip (diameter at the bottom: 350 μm, height: 450 μm) integrated into a needle, and the newly designed mold can fabricate an MN patch (arrays: 10 × 10). The distance between two MN was set to be 800 μm. Slow-dissolving mHA and fast-dissolving PVA/PVP were selected as the fillings of needle tips and needle bodies, respectively, to make MN separable. Once the prepared MN are inserted into the skin, the needle bodies rapidly dissolve, while the needle tips still maintain their structure, resulting in the sustained release of the drug loaded in the needle tips and the rapid release of the drug loaded in the needle bodies (Figure 1, the lower part).

Although HA has been reported as the filling of MN, HA-prepared MN usually lead to a rapid release of the loaded drug [35]. To slow down the drug release, methacrylate groups were introduced to the backbone of HA through an esterification reaction (Appendix A). The resulting polymer mHA contains double bonds, which can form a cross-linked network with the presence of light-inducer I 2959 under UV irradiation, thus sustaining the release of the loaded drug. The signals of the methacrylate proton and methyl proton on the ^1^H NMR spectra (Appendix A) and the characteristic absorption peaks of the ester group in FTIR (Appendix A) indicated the successful preparation of mHA. According to the peak area of methacrylate proton and methyl proton, the grafting rate was calculated to be ~70%.

MN were easily prepared by the micromolding method with three steps (Figure 1, the upper part): (i) loading mHA to fill the needle tips, (ii) loading PVA/PVP to fill the needle bodies, (iii) preparing the base layer with PVA/PVP, drying, demolding, and UV irradiation. The prepared MN were uniform with a smooth surface. A patch of MN contained 100 needles (10 × 10) with an intact base layer (Figure 2A). A cylindrical needle body and a conical needle tip integrated into a microneedle, which completely matched the shape of MN, as designed (Figure 2B and Appendix A). A fluorescence dye (EFL-DYE-UF-ENE-R) that can specifically bind to mHA was then used to visualize the distribution of mHA. The strong red fluorescent signal in Figure 2C confirmed that mHA mainly gathered in the needle tips and the size of the cone was in consistence with the designed needle tips. After water rinsing, the needle bodies were rapidly dissolved, while the needle tips still maintained their morphology with sharp tips, indicating that the prepared MN were separable, and they were able to release the loaded drugs with different speeds (Figure 2D and Appendix A).

### 3.2. Skin Insertion Capability of the Separable MN

The mechanical property of MN is an important parameter for their applications on transdermal drug delivery. Therefore, we first determined the mechanical strength of the designed separable MN by a texture analyzer to predict the possibility for skin insertion. As seen in Figure 3A, the curves of Force/Needle vs. Distance gradually climbed at the beginning and suddenly dropped when a force of 4~5 N was applied, indicating that MN bent [36]. MN without UV irradiation bent when a force of ~4 N was applied, while the force needed to be enhanced to 5 N to bend/break UV cured MN, indicating that UV curing was conducive to the mechanical properties of MN [37]. Compared with our previous work (force: 0.58 N), the newly designed MN with cones as needle tips and cylinders as needle bodies have great mechanical properties (force: 5 N) [20]. It was reported that a force of 0.1 N is sufficient to insert a needle into the skin [38]. Therefore, the prepared MN are supposed to be strong enough for skin insertion. As UV curing for 30 s yielded MN with a well sustained release of the loaded BSA-FITC (Figure 4A), we then evaluated the skin insertion capacity of the MN with UV irradiation for 30 s.

We next directly inserted the separable MN on the mouse skin, and trypan blue was used to visualize microholes in the skin. As presented in Figure 3B, the arrays of MN was clearly observed on the mouse skin after application of the MN on the mouse skin for 5 min, confirming that the obtained MN was strong enough to pierce the stratum corneum [39]. With the help of H & E staining, we observed that MN penetrated the skin to a depth of ~500 μm (Figure 3C), which is much deeper than that of general conical MN, resulting in higher drug delivery to the dermis of the skin without contacting the nerve fibers [36]. The incomplete insertion of MN attributes to the deformation of the skin during insertion [40]. Moreover, the skin gradually recovered after being punctured within 2 h, which demonstrated that the prepared MN were of excellent biocompatibility, showing no significant irritation to the skin (Figure 3D). The biosafety of separable MN is comparable to the mHA prepared MN (Appendix A). Apart from the normal camera, skin recovery can be carefully monitored with advanced technology, such as SEM, which supplies the pore closing process [41].

These results demonstrate that the separable MN with a novel texture designed in this study exert superior mechanical properties that enable efficient drug delivery. Moreover, the excellent biocompatibility of the separable MN ensures that MN can be used for further in vivo studies.

### 3.3. Drug Release from MN

To specifically explore the effect of UV curing on drug release from the needle tips, BSA-FITC was selected as a model drug and then prepared with the same procedure in the previous publication [42]. As shown in Figure 4A and Appendix A, mHA prepared MN without UV curing released ~100% of the drug within 20 min, which exhibited the same release character as the MN prepared from HA. As expected, UV curing slowed the release of BSA-FITC from the needle tips, indicating the successful crosslinking of mHA during UV irradiation. Furthermore, the longer the UV irradiation, the slower the drug release. We suspected that a denser network was formed with longer UV curing, thus delaying drug release. However, only ~75% of drug was released in 8 h when mHA prepared MN was cured with UV for 120 s. Given that MN prepared from mHA with UV curing for 30 s controlled BSA-FITC release in a sustained manner and enabled ~100% release of the loaded drug within 8 h, UV irradiation for 30 s was selected for the preparation of MN for the following research.

TCZ and Apt1-67 were then loaded in the needle tips and needle bodies to achieve a combined medication, respectively. Before preparing the dual-drug loaded MN, we first used agarose gel electrophoresis and CD spectroscopy to confirm that UV irradiation did not affect the structure of TCZ and Apt1-67. As shown in Figure 4B and Appendix A, all TCZ samples exhibited very similar CD spectra, and the same zeta potential was maintained after UV irradiation, indicating that UV curing had no significant influence on the structure of TCZ. Moreover, the band did not dash with the UV irradiation for various time periods, indicating that Apt1-67 maintained their structure with exposure to UV light, even for 120 s (Figure 4C). As MN were obtained in a dry status, the stability of dual-drug loaded MN can be deduced from the stability of dry DNA aptamer and dry TCZ.

TCZ and Apt1-67 were then mixed with mHA and PVA/PVP, respectively, to prepare the dual-drug loaded separable MN. The loading capacity of TCZ and Apt1-67 were determined to be 40.81 μg/patch and 1.33 μg/patch, respectively. With the help of the fluorescence labeling, we observed that TCZ-FITC mainly localized in the needle tips and Apt1-67-cy5 distributed in the needle bodies (Figure 4D). It is worth noting that some of Apt1-67-cy5 or free cy5 penetrated to the needle tips when loading Apt1-67-cy5 to the needle bodies, resulting in the presence of the signal of cy5 in the needle tips. TCZ-FITC and Apt1-67-cy5 loaded MN were then inserted into the mouse skin for 2 h to briefly investigate the drug’s release in vivo, and the skin was removed for ex vivo examination after the mouse was euthanized. As presented in Figure 4E, the overview of MN in the skin is much bigger than MN in a dry status (Figure 2B,C). The diameters of the needle tips and the needle bodies were estimated to be 540 μm and 700 μm, respectively, and the depth was approximately 600 μm, which was much deeper than the result determined in Figure 3C, indicating the swelling of MN and the release of the fluorescent drugs to the surroundings. With these comprehensive investigations, we concluded that TCZ and Apt1-67 loaded MN were successfully prepared according to the above-mentioned procedure without influencing their activities, and the loaded drugs were able to release to the surroundings after insertion into the skin for 2 h.

### 3.4. Anti-RA Efficacy of MN In Vivo

Encouraged by the excellent drug loading performance, skin insertion capability, and drug release property in vitro, we further examined the anti-RA activity of the developed dual-drug loaded separable MN against CIA mice. Before the anti-RA study, the cytotoxicity of MN was firstly evaluated using L929 and FLS as model cells, and the results demonstrated that both MN and dual-drug loaded MN had no significant toxicity to these cells (Appendix A), indicating that the developed separable MN were of great biocompatibility.

We further examined the anti-RA activity of dual-drug loaded MN against CIA mice (Figure 5A). After establishment of the CIA model, the CIA mice were grouped randomly and administrated every two days with TCZ/Apt1-67@MN, TCZ@MN, Apt1-67@MN, TCZ/Apt1-67 (SC), and saline, respectively. The paw thickness and the clinical scores were recorded every two days. After 20 days of treatment, the paw thickness of the mice treated with TCZ@MN and Apt1-67@MN decreased slightly, while TCZ/Apt1-67@MN and TCZ/Apt1-67 (SC) significantly reduced the paw thickness of the CIA mice (Figure 5B,C). Furthermore, compared with the models, the clinical score of the mice obviously dropped after different treatments. Inspiringly, the mice treated with TCZ/Apt1-67@MN or TCZ/Apt1-67(SC) scored less than the mice treated with a single drug loaded MN (TCZ@MN or Apt1-67@MN), revealing the better anti-RA activity of the combination of TCZ and Apt1-67 (TCZ/Apt1-67@MN or TCZ/Apt1-67(SC) (Figure 5D). These results demonstrated that the administration routes (transdermal administration via MN or SC) did not significantly affect the apparent symptoms of CIA mice and a combined medication exerted a better anti-RA activity. In addition, the treatments did not cause significant damage to the examined organs, indicating that the developed dual-drug loaded separable MN were of great biosafety (Appendix A).

### 3.5. Influences on Bone Structures and Cytokines

Bone erosion and synovial inflammation are prominent features of RA, and they are closely related to the severity of the inflammation [43]. Therefore, we further examined the pathology of the bone structure and the ankle joints after the treatment. Micro-CT is an X-ray imaging method that visualizes bone on a microscopic scale, and it can effectively and accurately assess bone structure [44]. As seen from Figure 6A, the bone of the model mouse characterized with a rough surface, joint deformities, and inflammatory erosion. The symptoms were not obviously improved with the treatment of TCZ@MN and Apt1-67@MN. On the contrary, the bones got smoother, with less inflammatory erosions after the treatment with TCZ/Apt1-67@MN and TCZ/Apt1-67 (SC). Furthermore, compared with the bone structures of CIA mice treated with TCZ/Apt1-67 (SC), the surface of CIA mice treated with TCZ/Apt1-67@MN were smoother, the boundaries at the tarsus were clearer, and the inflammatory erosion at the joints were obviously diminished, indicating that transdermal administration of TCZ and Apt1-67 via MN (TCZ/Apt1-67@MN) was more efficient against bone erosion than SC injection.

We next investigated the effects of different treatments on the ankle joints of the mice. The joint cavity of the mice treated with TCZ@MN and Apt1-67@MN exerted a significant synovial hyperplasia and intensive inflammatory cell infiltration, which were comparable with the models without any treatment (Figure 6B). In contrast, the mice in the TCZ/Apt1-67@MN and the TCZ/Apt1-67 (SC) groups exhibited less blood vessels and inflammatory cell infiltration (Figure 6B). It was worth noting that the symptoms, including synovial hyperplasia, inflammatory cell infiltration, and joint cavity roughness, were significantly relieved in mice treated with TCZ/Apt1-67@MN, demonstrating that TCZ/Apt1-67@MN were superior to TCZ/Apt1-67 (SC) in alleviating the joint symptoms of the CIA mice.

The drugs, TCZ and Apt1-67, loaded in the separable MN exhibit anti-RA activities by inhibiting IL-6R and TNFR1, and IL-6 and TNF-α are representative cytokines in these two signaling pathways. Therefore, the concentration of IL-6 and TNF-α in serum was determined to reflect the effects of TCZ/Apt1-67@MN on IL-6R- and TNFR1-mediated signaling pathways as well as the development of RA in mice. As seen from Figure 6C,D and Appendix A, all the treatments were able to reduce serum IL-6 and TNF-α, particularly in the TCZ/Apt1-67@MN and the TCZ/Apt1-67 (SC) groups. The inhibition rate of IL-6 of the TCZ/Apt1-67@MN and the TCZ/Apt1-67 (SC) groups were ~57% and ~64%, respectively. While the inhibition rate against TNF-α by TCZ/Apt1-67@MN and TCZ/Apt1-67(SC) were ~40% and ~47%, respectively. These results revealed that the combination of TCZ and Apt1-67 was more beneficial in reducing the cytokine level than a single administration. As proinflammatory factors, high levels of IL-6 and TNF-α exacerbate inflammation. The inhibition of IL-6 and TNF-α demonstrated that TCZ/Apt1-67@MN and TCZ/Apt1-67(SC) prevented the secretion of these two proinflammatory factors, speculating that TCZ and Apt1-67 inhibited inflammation by interfering with IL-6R- and TNFR1-mediated inflammatory pathways.

## 4. Conclusions

In summary, we successfully prepared dual-drug loaded separable MN with a novel texture in the present work. Slow-dissolving mHA and rapid-dissolving PVA/PVP were utilized as the fillings of needle tips and needle bodies, respectively, enabling the sustained release of TCZ and the fast release of Apt1-67. The prepared MN were of great mechanical property in the insertion of the mouse skin without inducing any irritation to the skin. The TCZ and the aptamer Apt1-67 distributed in the needle tips and the needle bodies as expected, and they were able to be released in the surroundings of the mouse skin. Compared with TCZ@MN and Apt1-67@MN, the dual-drug loaded MN (TCZ/Apt1-67@MN) exerted better efficacy. Moreover, MN-based transdermal delivery of TCZ and Apt1-67 was superior to SC injection in inhibiting bone erosion and relieving the symptoms of the joint cavity of the CIA mice. The present work not only provides a novel approach for a combined medication with different release properties but also provides a strategy for improving the efficacy of drugs.

## Figures and Tables

**Figure 1 pharmaceutics-14-01518-f001:**
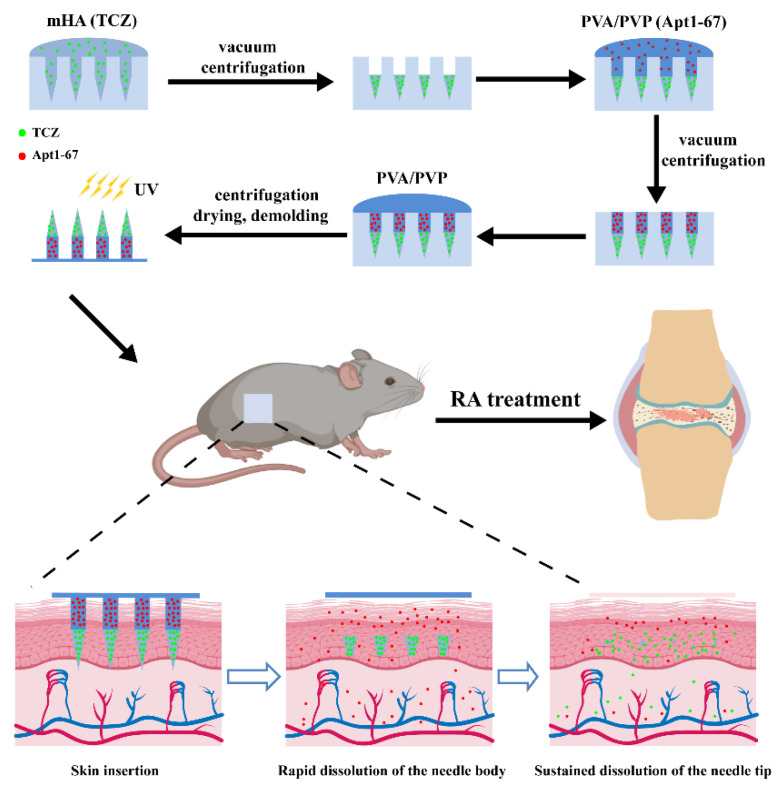
Schematic illustration of the preparation and the possible anti-RA activity of the dual-drug loaded separable MN in vivo. TCZ and Apt1-67 were loaded in the needle tips and needle bodies, respectively. After application of TCZ/Apt1-67@MN, the rapid dissolution of the needle bodies leads to the fast release of Apt1-67, while the slow dissolution of the needle tips enables a sustained release of TCZ. The released drugs directly enter into the blood circulation, resulting in a combined medication of Apt1-67 and TCZ for efficient RA therapy.

**Figure 2 pharmaceutics-14-01518-f002:**
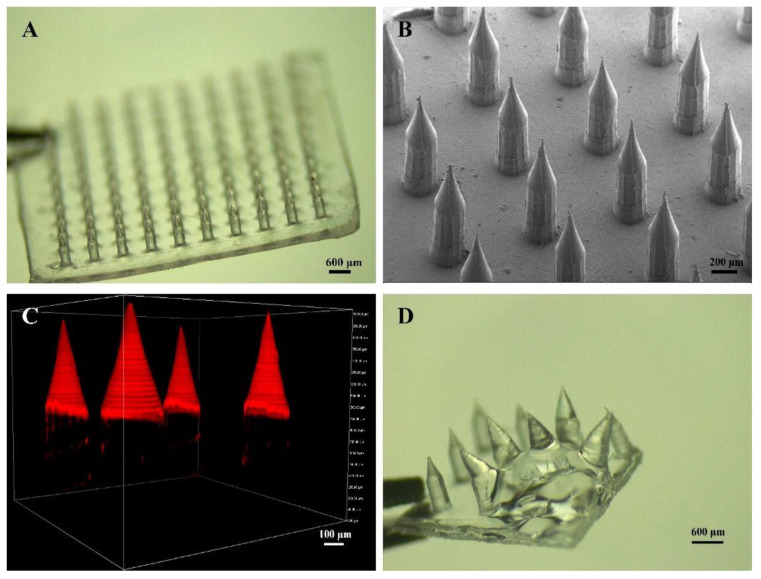
Characterization of the separable MN. (**A**) Optical image. (**B**) SEM image. (**C**) 3D image of the separable MN prepared by EFL-DYE-UF-ENE-R labeled mHA. (**D**) Optical image of the separable MN after water rinsing.

**Figure 3 pharmaceutics-14-01518-f003:**
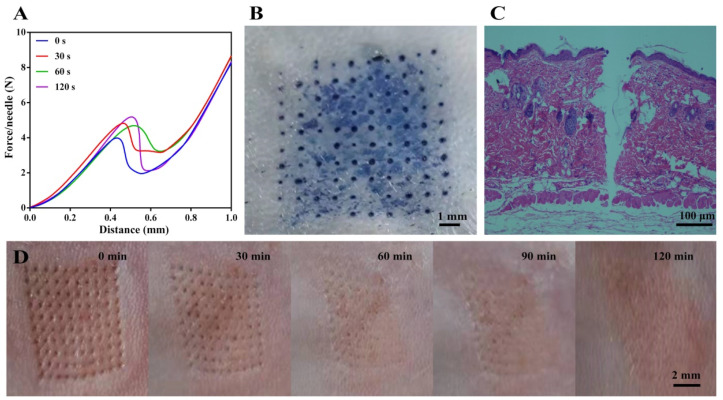
Mechanical property of the separable MN. (**A**) Average mechanical strength of single microneedle. (**B**) Mouse skin insertion with trypan blue staining. (**C**) H & E staining of mouse skin after MN application. (**D**) Time-dependent skin recovery after MN insertion.

**Figure 4 pharmaceutics-14-01518-f004:**
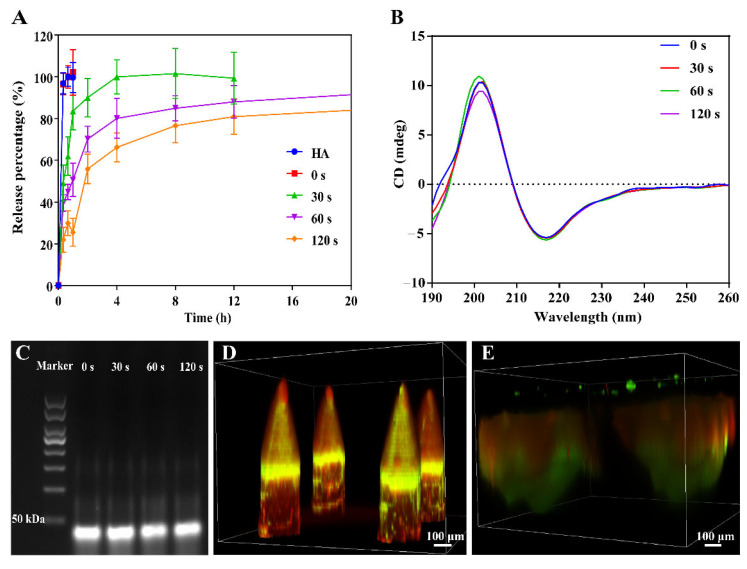
Drugs loading and release. (**A**) Release of model drug BSA-FITC from the needle tips of the separable MN. (**B**) CD spectra of TCZ after UV irradiation. (**C**) Agarose gel electrophoresis of Apt1-67 after UV irradiation. (**D**) CLSM image of TCZ-FITC and Apt1-67-cy5 loaded separable MN. (**E**) Ex vivo examination of the mouse skin after application of TCZ-FITC and Apt1-67-cy5 loaded MN.

**Figure 5 pharmaceutics-14-01518-f005:**
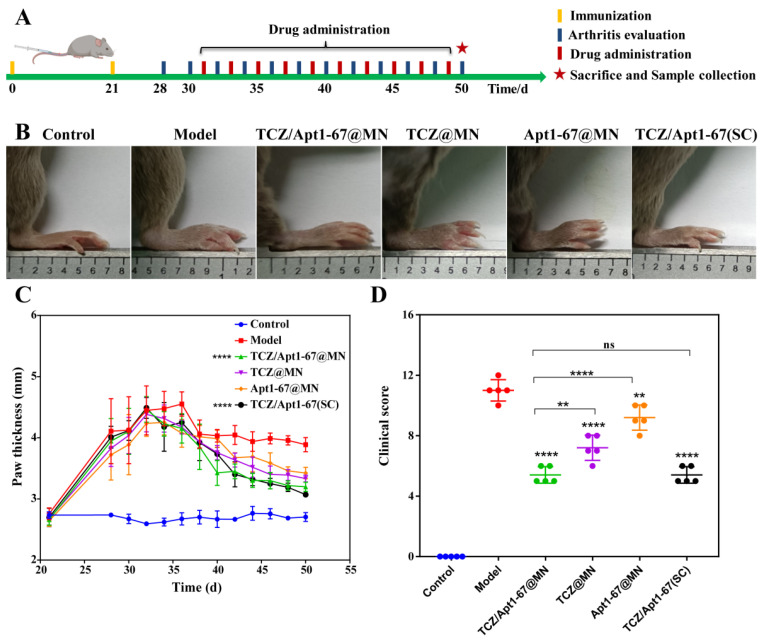
Anti-RA activity of the separable MN in vivo. (**A**) Outline of the in vivo experiments. (**B**) Images of the paws of the mice after various treatments. (**C**) Time-dependent paw thickness with various treatments. (**D**) Clinical score of the mice after various treatments. **: *p* < 0.01; ****: *p* < 0.0001; ns: not significant.

**Figure 6 pharmaceutics-14-01518-f006:**
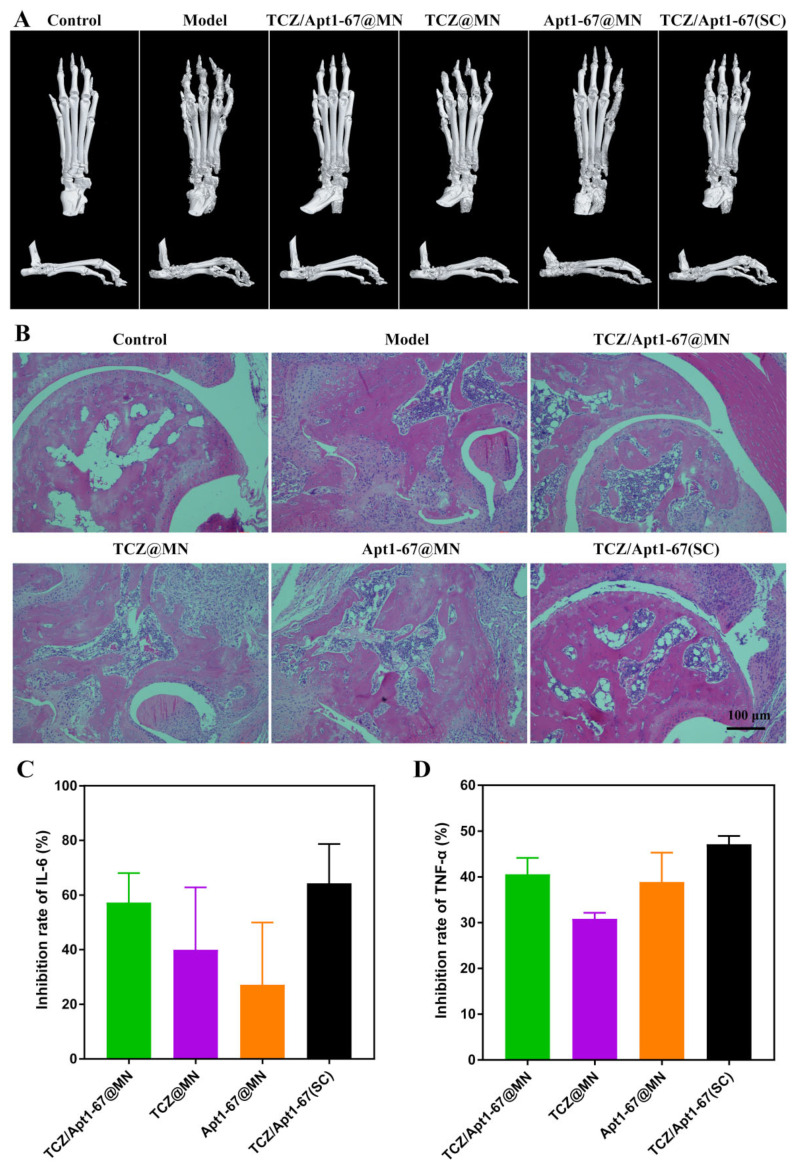
The effects on bone structures and cytokines. (**A**) Micro-CT of ankle joints. (**B**) H & E staining of the ankle joints. (**C**) Inhibition of IL-6 and (**D**) TNF-α by the various treatments.

## Data Availability

The data presented in this study are available on request from the corresponding author.

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
