# Peer review of "Dual-Drug Loaded Separable Microneedles for Efficient Rheumatoid Arthritis Therapy"

_pharmaceutics, 2022, doi:10.3390/pharmaceutics14071518_

Round 1

Reviewer 1 Report

Dear authors,

In general, this work is very interesting and novel and well presented and surely it is worth publication.

The introduction summarizes the prior studies about the topic in a manner that lays a foundation for understanding the research problem and it addresses the gaps in the literature and clearly stats the purpose of the work. The experimental part is well-designed, and you have used appropriate instrumentations/techniques for characterization and testing. The materials and methods section is illustrative. The results are clearly presented and sufficiently discussed and finally, the conclusions are well supported by the results.

There are a few minor corrections and some typos need fixing.

1- Page 6, line 239: Please add this schematic diagram to the figures and label it as Figure 1. 

2- In the schematic diagram: replace the word solution with hydrogel in three labels: mHA solution (TCZ),  PVA/PVP (APT1-67) and PVA/PVP solution.

Thank you

Ismaiel

Author Response

Comments

In general, this work is very interesting and novel and well presented and surely it is worth publication.

The introduction summarizes the prior studies about the topic in a manner that lays a foundation for understanding the research problem and it addresses the gaps in the literature and clearly stats the purpose of the work. The experimental part is well-designed, and you have used appropriate instrumentations/techniques for characterization and testing. The materials and methods section is illustrative. The results are clearly presented and sufficiently discussed and finally, the conclusions are well supported by the results.

There are a few minor corrections and some typos need fixing.

  1. Page 6, line 239: Please add this schematic diagram to the figures and label it as Figure 1.

Our reply: Thanks for your kind suggestion! Scheme 1 has been labeled as Figure 1.

  1. In the schematic diagram: replace the word solution with hydrogel in three labels: mHA solution (TCZ), PVA/PVP (APT1-67) and PVA/PVP solution.

Our reply: The figure has been revised.

Reviewer 2 Report

The manuscript was well done, showing merit to be approved without corrections. The paper showed an innovative idea about microneedle and presented excellent results. I agree with the structure, results, and discussion. The use of dual drugs is a future of the treatment efficiently and quickly.

Author Response

Comments

The manuscript was well done, showing merit to be approved without corrections. The paper showed an innovative idea about microneedle and presented excellent results. I agree with the structure, results, and discussion. The use of dual drugs is a future of the treatment efficiently and quickly.

Our reply: Thank you very much for your positive comments! We are highly encouraged to continue the other side-projects of mincroneedles.

Reviewer 3 Report

The idea of transdermal delivery of dual RA alleviating drug through microneedles sounds promising. Their paper presented data on how to extend the release of RA drug overtime by introducing a slow degrading polymer. To use a patch to deliver a cocktail of therapeutic drug is always significant as compared to SC or IV injection. They have shown, through their preclinical data, that drug loaded patch attached on skin showed similar effects to SC injection. Therefore, their work has both novelty and the originality. And since UV curing the MN body with embedded drug doesn’t affect the integrity of the drug (CD and gel electrophoresis data), the work they are doing to deliver it through skin is important.

However, some comments to be addressed.

·         The authors didn’t perform a transverse fracture test to show the MN fracture force. This is quite important in dual layered MNs, this determines at what force does the 1st layer (MN tip) will break away from the 2nd layer, or whether that occurs before the penetration event.

·         There were few gaps and inconsistencies in presenting the data.

·         (1) Section 2.3 (Fabrication of the separable MN) doesn’t present the geometric dimensional of the PDMS mold. How were the mold prepared?

·         It is mentioned that PDMS was bought, but no mention of how the PDMS mold were created.

·         (2) Section 2.4 (characterization of MN) shows axial force vs. displacement of the needle body. Since there are two layers, should a transverse force also be measured?

·         Section 3.1, line 263, uses the word ‘only gathered’ in the context of mHA labelled dye being concentrated in the tip, which could imply that 100% of mHA is in the tip, however, Fig.1C shows some mHA-dye intensity in the body of needle (very few red lines), which is indicative of two polymer mixture infiltrating into each other.

·         Fig.1D shows the image of MN after rinsing with water. The body is made of PVA/PVP. If the body dissolves quickly, then the tips with dislodge (mHA has longer degradation rate), but the figure, instead, shows swelling of the body while the tips are still attached. Please explain?

·         Fig. 2C shows the histological image of the one penetration site. Could it be a histological artifact? During the sectioning stage, did the skin tear?

·         Line 337-341 didn’t make sense since the authors are claiming that, according to Fig. 3e, the diameter of the penetration site and the penetration depth are both greater than the actual dimensions of the MNs (Fig. 1B-C). They have compared the dye from the MNs diffusion into the surrounding tissue (CLSM, Fig. 3E) to the dimensions of MN body (pre-insertion).

·         Wouldn’t the pores start to close straight away as demonstrated in (https://doi.org/10.1016/j.jconrel.2019.05.024) Please comment on this and add relevant literature such as this paper.

·         The intext referencing styles for figures are a little confusing. For example, section 3.4 is data rich, the two figures (4 and 5) contain data that are condensed into categories, but the categories are not sub-numbered. It is a little difficult to track which sub-figure they are referring to when reading a paragraph.

·         (13) Fig.1A and D- check the image scale bar. Both scale bars show similar length of 600 µm. The needles look bigger in Fig.1D than Fig.1A as a digital zoom might have been applied. The scale bar needs to be rescaled.

Author Response

Comments

The idea of transdermal delivery of dual RA alleviating drug through microneedles sounds promising. Their paper presented data on how to extend the release of RA drug overtime by introducing a slow degrading polymer. To use a patch to deliver a cocktail of therapeutic drug is always significant as compared to SC or IV injection. They have shown, through their preclinical data, that drug loaded patch attached on skin showed similar effects to SC injection. Therefore, their work has both novelty and the originality. And since UV curing the MN body with embedded drug doesn’t affect the integrity of the drug (CD and gel electrophoresis data), the work they are doing to deliver it through skin is important.

However, some comments to be addressed.

  1. The authors didn’t perform a transverse fracture test to show the MN fracture force. This is quite important in dual layered MNs, this determines at what force does the 1st layer (MN tip) will break away from the 2nd layer, or whether that occurs before the penetration event.

Our reply: Thank you very much for your kind suggestion! Separation of the two layers was not observed before the penetration by naked ayes. Instead of the transverse fracture test, the force that bends MN was tested by a texture analyzer. Deep discussion and citations have been added in the manuscript.

As seen in Figure 3A, the curves of Force/Needle vs. Distance gradually climbs at the beginning and suddenly dropped when a force of 4~5 N was applied, indicating that MN bent. The force that bends or breaks MN was far away from the force (0.1 N) that needed to insert MN in the skin.

Determination of the fraction force would supply deeper information of MN. However, we have not found a suitable instrument to get such parameter. Given that a transverse force rarely involves in the application of MN, fracture force was not supplied.

  1. There were few gaps and inconsistencies in presenting the data.

Our reply: Thanks for your comments! Corrections have been made and can be tracked in the main text.

  1. Section 2.3 (Fabrication of the separable MN) doesn’t present the geometric dimensional of the PDMS mold. How were the mold prepared? It is mentioned that PDMS was bought, but no mention of how the PDMS mold were created.

Our reply: Thank you very much! A schematic illustration of the PDMS mold has been included in Figure S1A.

To prepare a PDMS mold, a master mold should be first fabricated. A mixture of liquid PDMS and curing agent is then poured into the master mold. After drying, the raw PDMS mold can be peeled off and clipped to obtain the final PDMS.

However, fabrication the master mold is another subject which is very challenging for all co-authors. Therefore, we designed the MN arrays and a professional company (Taizhou Microchip Pharmaceutical Technology Co., Ltd.) provided us the service to produce a master mold and we bought PDMS mold from the company. As the fabrication of the master mold is confidential, we were not supplied the details of the fabrication of the master mold.

  1. Section 2.4 (characterization of MN) shows axial force vs. displacement of the needle body. Since there are two layers, should a transverse force also be measured?

Our reply: Thank you for the valuable suggestion! We would like to refer to the answer to question 1.

To punch MN into the skin, an axial force is normally applied. Therefore, the axial force that bends the developed MN was determined by a texture analyzer (as described in section 2.4 and Figure 3A). Measuring the transverse force is a great suggestion to better understanding the developed separable MN. However, we have not found a suitable instrument to determine the transverse force, and a transverse force rarely involves in the application of MN. Hence, a transverse fracture test is not included.

  1. Section 3.1, line 263, uses the word ‘only gathered’ in the context of mHA labelled dye being concentrated in the tip, which could imply that 100% of mHA is in the tip, however, Fig.1C shows some mHA-dye intensity in the body of needle (very few red lines), which is indicative of two polymer mixture infiltrating into each other.

Our reply: Thanks for pointing out the misnomer! We have revised this sentence as follows:

The strong red fluorescent signal in Figure 2C confirmed that mHA mainly gathered in the needle tips and the size of the cone was in consistence with the designed needle tips.

  1. Fig.1D shows the image of MN after rinsing with water. The body is made of PVA/PVP. If the body dissolves quickly, then the tips with dislodge (mHA has longer degradation rate), but the figure, instead, shows swelling of the body while the tips are still attached. Please explain?

Our reply: A detailed separation process of MN was included in Figure S3.

As described in the Section 2.3., MN were obtained after drying. Three processes may involve in MN dissolving: (i) water diffusing into the matrix of MN, (ii) MN swelling, and (iii) dissolving. As seen in Figure 2D, the needle bodies swelled quickly when rinsed with water. With longer incubation, the needlebody dissolved (Figure S3B) and the needletips slipped into the water (Figure S3C). These results fully support the conclusion that the prepared MN are separable.

  1. Fig. 2C shows the histological image of the one penetration site. Could it be a histological artifact? During the sectioning stage, did the skin tear?

Our reply: The authors highly disagree the doubt “a histological artifact”. Figure 3C was included in the manuscript as a representative. More images are attached for your kind references. To make sure the results were reproducible, MN from different batches were applied to the mouse skin.

H&E staining of mouse skin after MN application. Please note that image (A) and image (B) are obtained with different batches of MN. Scale bar: (A) 100 μm, (B) 600 μm.

After MN application, the skin was removed from the body, fixed with tissue fixation, embedded in paraffin, sectioned, stained with hematoxylin and eosin (H&E) (section 2.4, lines 147~149). The sample was solid while sectioning. Therefore, no liquid was observed.

  1. Line 337-341 didn’t make sense since the authors are claiming that, according to Fig. 3e, the diameter of the penetration site and the penetration depth are both greater than the actual dimensions of the MNs (Fig. 1B-C). They have compared the dye from the MNs diffusion into the surrounding tissue (CLSM, Fig. 3E) to the dimensions of MN body (pre-insertion).

Our reply: Thank you very much for your comment! We have made corrections in the manuscript accordingly.

Fluorescence labeled drugs were used to visualize the drug distribution throughout MN in dry status (Figure 4D). As the answer to question 5, the three processes may involve in MN dissolving: (i) water diffusing into the matrix of MN, (ii) MN swelling, and (iii) dissolving. Therefore, increase in the dimensions of MN in Figure 4E (Figure 3E in the original version) was the results of both MN swelling and drug diffusion.

  1. Wouldn’t the pores start to close straight away as demonstrated in (https://doi.org/10.1016/j.jconrel.2019.05.024) Please comment on this and add relevant literature such as this paper.

Our reply: Thank you for your inspiring comments! Details of the pore closing process are quiet interesting. Discussion has been included in the manuscript and the paper mentioned here has been cited.

  1. The intext referencing styles for figures are a little confusing. For example, section 3.4 is data rich, the two figures (4 and 5) contain data that are condensed into categories, but the categories are not sub-numbered. It is a little difficult to track which sub-figure they are referring to when reading a paragraph.

Our reply: Thanks for your suggestion! We have divided this part into two sections: 3.4. Anti-RA efficacy of MN in vivo, and 3.5. Influences on bone structures and cytokines.

  1. Fig.1A and D- check the image scale bar. Both scale bars show similar length of 600 µm. The needles look bigger in Fig.1D than Fig.1A as a digital zoom might have been applied. The scale bar needs to be rescaled.

Our reply: We have carefully checked the original data and the scale bars have been rescaled.

Reviewer 4 Report

The author designed the dual-drug loaded separable microneedles for rheumatoid arthritis therapy, and evaluated the usefulness. Our comments are followings.

1) Figure 2D: Although, MN were inserted into the skin of the mouse for 10 min, the drug release observed approximately 4 h. Author should mention the effect of skin recovery after 4 h treatment. In addition, H&E staining of skin recovery in mouse skin 120 min after removing the MN also support the safety of MN application.

2) Figure 2D: More detailed safety checks are needed. If possible, please compare for the skin toxicity and skin recovery between of HA group and separable MN group.

3) Figure 4 and 5: Author should discuss whether the anti-RA activity of the separable MN is higher than that in vehicle and HA group. This comparison is important to show the usefulness of separable MN.

4) What is the expiration date of the product after preparation? Please mention the degradation stability of the drug in the separable MN.

5) Figure 5C and D: Please show the data of IL-6 and TNF-α in the normal and vehicle-treated model (or non-treatment model).

Author Response

Comments

The author designed the dual-drug loaded separable microneedles for rheumatoid arthritis therapy, and evaluated the usefulness. Our comments are followings.

  1. Figure 2D: Although, MN were inserted into the skin of the mouse for 10 min, the drug release observed approximately 4 h. Author should mention the effect of skin recovery after 4 h treatment. In addition, H&E staining of skin recovery in mouse skin 120 min after removing the MN also support the safety of MN application.

Our reply: Thanks for your comments! Figure 2D shows the skin recovery after application of MN for 10 min, which primarily confirmed the biocompatibility of MN for further in vivo study.

To examine the status of the skin after application of MN for 4 h, HE staining was performed and images were taken. After 4 h, the microholes were closed from the outside (A). Similar results were observed after skin recovery for 2 h (B).

HE staining of mouse skin after application of MN for 4 h (A) and recovery for 2 h (B).

Furthermore, the histomorphology of major organs after therapy was investigated. Compared with the control group, all the treatments did not cause significant damages to the examined organs, indicating that the developed dual-drug loaded separable MN were of great biosafety (Figure S8).

  1. Figure 2D: More detailed safety checks are needed. If possible, please compare for the skin toxicity and skin recovery between of HA group and separable MN group.

Our reply: Skin recovery after application of mHA prepared MN has been included in Figure S4. The results demonstrated that the biosaftety of mHA MN is comparable to that of separable MN.

  1. Figure 4 and 5: Author should discuss whether the anti-RA activity of the separable MN is higher than that in vehicle and HA group. This comparison is important to show the usefulness of separable MN.

Our reply: Thank you very much for your suggestion! The goal of this work was to develop separable MN as a co-delivery system for RA therapy. The separability endowed the developed MN to release the loaded drug with different properties. Rapid dissolution of the needle body (filling: PVA/PVP) led to fast release of Apt1-67 and separation of the needle tips (filling: mHA) from needle bodies. Since the developed MN were developed as a co-delivery system, single drug loaded MN (TCZ@MN and Apt1-67@MN) were used instead of separable MN as controls for the study of anti-RA activity.

  1. What is the expiration date of the product after preparation? Please mention the degradation stability of the drug in the separable MN.

Our reply: Honestly, this is a fundamental research and we have not tested the expiration date of the developed MN. MN were freshly prepared and used for both in vitro and in vivo experiment. As MN were obtained in dry status, the stability of dual-drug loaded MN can be deduced from the stability of dry DNA aptamer and dry TCZ. An extensive study will be performed on the stability of MN as well as storage conditions for new drug development.

  1. Figure 5C and D: Please show the data of IL-6 and TNF-α in the normal and vehicle-treated model (or non-treatment model).

Our reply: The serum IL-6 of the mice in the group of control, model, TCZ/Apt1-67@MN, TCZ@MN, Apt1-67@MN, and TCZ/Apt1-67 (SC) was 4.9, 10.3, 4.3, 6.0, 7.4, and 3.7 pg/mL, respectively. The serum TNF-α of the mice in the group of control, model, TCZ/Apt1-67@MN, TCZ@MN, Apt1-67@MN, and TCZ/Apt1-67 (SC) was 102.4, 212.4, 137.1, 154.7, 147.0, and 125.7 pg/mL, respectively. Inhibition rate would better reflect the influences of the list therapeutics on the cytokines.

Round 2

Reviewer 4 Report

Our comments are followings.

What is the expiration date of the product after preparation? Please mention the degradation stability of the drug in the separable MN.

Our reply: Honestly, this is a fundamental research and we have not tested the expiration date of the developed MN. MN were freshly prepared and used for both in vitro and in vivo experiment. As MN were obtained in dry status, the stability of dual-drug loaded MN can be deduced from the stability of dry DNA aptamer and dry TCZ. An extensive study will be performed on the stability of MN as well as storage conditions for new drug development.

Reviewer: Thank you. Please mention this contents in the discussion.

Figure 5C and D: Please show the data of IL-6 and TNF-α in the normal and vehicle-treated model (or non-treatment model).

Our reply: The serum IL-6 of the mice in the group of control, model, TCZ/Apt1-67@MN, TCZ@MN, Apt1-67@MN, and TCZ/Apt1-67 (SC) was 4.9, 10.3, 4.3, 6.0, 7.4, and 3.7 pg/mL, respectively. The serum TNF-α of the mice in the group of control, model, TCZ/Apt1-67@MN, TCZ@MN, Apt1-67@MN, and TCZ/Apt1-67 (SC) was 102.4, 212.4, 137.1, 154.7, 147.0, and 125.7 pg/mL, respectively. Inhibition rate would better reflect the influences of the list therapeutics on the cytokines.

Reviewer: Thank you. Please add these data in the result.

Author Response

Dear reviewer,

Thank you very much for your kind suggestions! I believe that our manuscript benefits a lot from your comments. Our replies are attached as follows:

#Q1: What is the expiration date of the product after preparation? Please mention the degradation stability of the drug in the separable MN.

Our reply: Honestly, this is a fundamental research and we have not tested the expiration date of the developed MN. MN were freshly prepared and used for both in vitro and in vivo experiment. As MN were obtained in dry status, the stability of dual-drug loaded MN can be deduced from the stability of dry DNA aptamer and dry TCZ. An extensive study will be performed on the stability of MN as well as storage conditions for new drug development.

Reviewer: Thank you. Please mention this contents in the discussion.

Our reply: Thanks for your suggestion! Discussion on the degradation stability of the drug in MN has been added in the main text (lines 338-340).

#Q2: Figure 5C and D: Please show the data of IL-6 and TNF-α in the normal and vehicle-treated model (or non-treatment model).

Our reply: The serum IL-6 of the mice in the group of control, model, TCZ/Apt1-67@MN, TCZ@MN, Apt1-67@MN, and TCZ/Apt1-67 (SC) was 4.9, 10.3, 4.3, 6.0, 7.4, and 3.7 pg/mL, respectively. The serum TNF-α of the mice in the group of control, model, TCZ/Apt1-67@MN, TCZ@MN, Apt1-67@MN, and TCZ/Apt1-67 (SC) was 102.4, 212.4, 137.1, 154.7, 147.0, and 125.7 pg/mL, respectively. Inhibition rate would better reflect the influences of the list therapeutics on the cytokines.

Reviewer: Thank you. Please add these data in the result.

Our reply:  The concentration of cytokines IL6 and TNF-α as Figure S9 in the supplementary materials.

Thank you again for your effort!

Kind regards,

Yongxing Zhao & Pengchao Sun